# Targeted-Gene Sequencing to Catch Triple Negative Breast Cancer Heterogeneity before and after Neoadjuvant Chemotherapy

**DOI:** 10.3390/cancers11111753

**Published:** 2019-11-08

**Authors:** Serena Di Cosimo, Valentina Appierto, Marco Silvestri, Giancarlo Pruneri, Andrea Vingiani, Federica Perrone, Adele Busico, Secondo Folli, Gianfranco Scaperrotta, Filippo Guglielmo de Braud, Giulia Valeria Bianchi, Stefano Cavalieri, Maria Grazia Daidone, Matteo Dugo

**Affiliations:** 1Biomarker Unit, Department of Applied Research and Technological Development, Fondazione IRCCS Istituto Nazionale dei Tumori di Milano, Via Giovanni Antonio Amadeo 42, 20133 Milano, Italy; Valentina.Appierto@istitutotumori.mi.it (V.A.); Marco.Silvestri@istitutotumori.mi.it (M.S.); Giancarlo.Pruneri@istitutotumori.mi.it (G.P.); Andrea.Vingiani@istitutotumori.mi.it (A.V.); Federica.Perrone@istitutotumori.mi.it (F.P.); Adele.Busico@istitutotumori.mi.it (A.B.); Secondo.Folli@istitutotumori.mi.it (S.F.); Gianfranco.Scaperrotta@istitutotumori.mi.it) (G.S.); Filippo.deBraud@istitutotumori.mi.it (F.G.d.B.); Giulia.Bianchi@istitutotumori.mi.it (G.V.B.); Stefano.Cavalieri@istitutotumori.mi.it (S.C.); MariaGrazia.Daidone@istitutotumori.mi.it (M.G.D.); 2Oncology and Hemato-Oncology Department, University of Milan, Via Festa del Perdono 7, 20122 Milano, Italy

**Keywords:** targeted-gene sequencing, heterogeneity, biopsies, neo-adjuvant chemotherapy, triple negative breast cancer, somatic mutations, cancer therapy

## Abstract

Triple negative breast cancer (TNBC) patients not attaining pathological Complete Response (pCR) after neo-adjuvant chemotherapy (NAC) have poor prognosis. We characterized 19 patients for somatic mutations in primary tumor biopsy and residual disease (RD) at surgery by 409 cancer-related gene sequencing (IonAmpliSeqTM Comprehensive Cancer Panel). A median of four (range 1–66) genes was mutated in each primary tumor biopsy, and the most common mutated gene was *TP53* followed by a long tail of low frequency mutations. There were no recurrent mutations significantly associated with pCR. However, half of patients with RD had primary tumor biopsy with mutations in genes related to the immune system compared with none of those achieving pCR. Overall, the number of mutations showed a downward trend in post- as compared to pre-NAC samples. *PIK3CA* was the most common altered gene after NAC. The mutational profile of TNBC during treatment as inferred from patterns of mutant allele frequencies in matched pre-and post-NAC samples showed that RD harbored alterations of cell cycle progression, PI3K/Akt/mTOR, and EGFR tyrosine kinase inhibitor-resistance pathways. Our findings support the use of targeted-gene sequencing for TNBC therapeutic development, as patients without pCR may present mutations of immune-related pathways in their primary tumor biopsy, or actionable targets in the RD.

## 1. Introduction

Breast cancer (BC) is a complex disease with different biological characteristics and clinical behavior. Triple-negative breast cancer (TNBC), defined by the absence of estrogen (ER), progesterone (PR), overexpression and/or gene amplification of HER2 receptors, accounts for approximately 15% of all BC cases [1]. Patients with TNBC typically have a dismal prognosis, due to an inherently aggressive clinical behavior, and a lack of recognized targeted therapy [1]. Hence, most patients with TNBC will be considered for chemotherapy at some point in their care [2,3,4]. Aside from the potential clinical benefits that are achieved by down-staging BC, neo-adjuvant chemotherapy (NAC) allows direct and early observation of efficacy of treatment, and could provide an opportunity to study the impact of treatment on TNBC biology. Furthermore, NAC provides important prognostic information since those patients with no evidence of invasive cancer in the breast and axillary nodes (pathological complete response (pCR)) have significantly improved long-term outcomes relative to patients with residual disease (RD) in surgical specimens [5]. Indeed, the current standard of post-surgical systemic treatment in patients with TNBC disease is still debated [6]. Hence, understanding the effect of NAC in clinical samples is critical to inform the design of combination or post-surgical sequential therapies. In addition, it has been proposed that a combination of chemotherapy and immune therapy may result in a supra-additive effect through multiple mechanisms, including increased mutational (and neoantigen) load induced by DNA-damaging agents, such as chemotherapeutic regimens [7]. As we move towards precision medicine, there is a growing interest in molecular classification of TNBC [8,9]. Besides, the definition of molecular changes within individual patients during treatment is increasingly appreciated to overcome the limits of a decision-making process merely based on basal tumor assessment [10,11]. Several studies on BC genomic profile as defined by whole exome sequencing (WES) revealed a considerable number of somatic mutations in tumor relevant genes [12,13,14]. Although these studies have expanded our knowledge on BC biology, they found a limit in the heterogeneity of the analyzed patient population for clinical stage, molecular intrinsic subtype and/or type of treatment as well as in the complexity of the method used, as WES implementation in routine diagnostics is currently prevented by cost, and time to results. Based on this premise, the goal of our study was to use a pragmatic approach with targeted-gene sequencing to analyze primary tumor somatic mutations and their changes in post-treatment samples of a homogenous cohort of TNBC patients.

## 2. Results

### 2.1. Patient Characteristics

A total of 19 patients scheduled to anthracycline/taxane-based NAC were included in this analysis. Relevant demographic, treatment, and tumor characteristics are summarized in Table 1. Median patient age was 43 years (range 32–75). Fifteen patients presented with stage II disease, three with stage III, and one patient with suspicious bone metastases at initial diagnosis. All cases but one were grade 3. After a median number of six (range 4–8) cycles of NAC, the pCR rate was 21%. With a median follow up of 70 months (95%CI 50–81), seven (37%) patients had relapsed. Pre- and post-NAC tumor sequencing was feasible in 16 (84%), and 15 (79%) of the 19 cases, respectively. Inadequate DNA content (pre-NAC cases = 3) and unavailability of post-NAC tumor tissue (in four cases achieving pCR) prevented mutation analysis in missing cases. For 12 (63%) patients both pre- and post-NAC tumor specimens were analyzed (Appendix A).

### 2.2. Somatic Mutations in TNBC Samples

Somatic mutations with a high or moderate impact on protein function, including single nucleotide variants (SNV), multiple nucleotide variants (MNV), and small insertions and deletions (INDEL) were detected in all 31 sequenced tumor samples, with a median of four mutations per sample (Figure 1A, Appendix A). The majority of detected mutations were missense, followed by truncating mutations and variants affecting splicing and the most frequent nucleotide variation was C > T/G > A (Figure 1A). Among the 10 most frequently mutated genes we found *TP53*, altered in 87% of pre-NAC, and in 60% of post-NAC samples, followed by a long tail of less recurrently mutated genes (Figure 1B).

To assess whether the mutational spectrum observed in our study was representative of TNBC we analyzed mutational data of TNBC patients of The Cancer Genome Atlas (TCGA) for the same 409 genes included in our panel. A similar pattern, in terms of mutation types, nucleotide change, number of mutations per patient, and recurrently altered genes was observed (Appendix A). A decreased number of mutations was observed in patients ≤ 50 compared to those >50 years old, 4 (2–6) vs. 7 (5–66), *p* = 0.002 (Appendix A). This association is unlikely confounded by other factors since the proportion of stage, grade, and type of treatment was similar among different age categories in this case series.

### 2.3. Genes and Pathways Associated with pCR

Considering pre-NAC samples, we first assessed whether patients with pCR or RD had a different tumor mutational load. No significant difference was observed in the number of mutations between the two groups, both considering mutations with high/moderate impact and overall detected somatic mutations. Furthermore, the frequency of mutated genes was similar in patients with and without pCR. Beside *TP53* that was mutated in 87% of pre-NAC tumors, most genes were mutated in few patients and no significant association between altered genes and NAC response was found. When we grouped mutated genes by pathway no significant association with pCR was observed (Appendix A). Nevertheless, 50% of patients with RD (*n* = 6) were characterized by alterations in pathways related to adaptive immunity, and in particular in B cell and T cell signaling, and in ERBB signaling (Figure 2A). None of the four patients attaining pCR showed alterations in these pathways. Beside those encoding for cell-surface receptors (KIT, FLT3, ERBB2), transmembrane proteins (GPR124 and TGFBR2), cytoplasmic proteins for signal transduction (*PIK3CA*, *KRAS*, MAP2K2, *MTOR*), and Janus family (JAK1, JAK3, and TYK2), protein trafficking and membrane integrity (TPR, NUP214, *CARD11*, ITGB2), extracellular matrix (FN1), antioxidant response (KEAP1), and transcription factors (REL), it is noteworthy the Inhibitor of nuclear factor kappa-B kinase subunit beta (*IKBKB*) gene encoding for a serine kinase that plays an essential role in the NF-kappa-B signaling pathway, and the RNASEL gene encoding for a component of the interferon-regulated 2-5A system that functions in the antiviral and antiproliferative roles of interferons. Several genes, such as *PIK3CA*, *KRAS*, *IKBKB*, *MTOR*, and *CARD11*, were recurrent across these pathways but each of them was mutated in only one or two patients, highlighting a highly individual heterogeneity of the mutational profiles (Figure 2B).

### 2.4. Comparison between Pre- and Post-NAC Samples

Comparison of the tumor mutational load between pre- (*n* = 16) and post-NAC (*n* = 15) samples showed no differences considering either high and moderate impact mutations, or all detected somatic mutations (Appendix A). Clustering of pre- and post-NAC tumor pairs according to VAF of somatic mutations showed that for some patients the mutational profiles of tumors were conserved during NAC, while for others pre- and post-NAC samples had different mutations (Appendix A). Examination of detected mutations showed that there was a portion of mutations private to (i.e., observed only in) pre- and post-NAC tumors; the median number of mutations private to pre- and post-NAC tumors was 2 (range 0–65) and 0.5 (range 0–3), respectively; there were very few shared mutations, and the median number of common mutations was 2 (range 0–10) (Appendix A).

To further assess the characteristics of genomic alteration in unresponsive TNBC patients, we examined individual cases evaluating the different mutant allele frequency distribution between pre- and post-NAC sample. In eight out of 12 (66%), i.e., patient (p)5, p6, p10, p11, p12, p13, p16, and p18 the mutant alleles frequencies differ between pre- and post-NAC samples, identifying groups of residual mutations. The representative case of this behavior, patient p16, is characterized by loss of 15 genes (Appendix A) in the post-NAC sample, and by the presence of residual mutations in cluster C2, represented by *SOX11*, *TAF1L*, and *TCF7L1* genes (Figure 3A). In four out of 12 (33%) patients, i.e., p1, p4, p14, and p15, the frequency of single mutations did not change between pre and post-NAC conditions. Considering p4 as a representative case, the highest VAF was detected for cluster C1 (Appendix A) involving *LTK*, *SF3B1*, *TP53* genes in both pre- and post-NAC samples (Figure 3B). Notably, all patients with stable clusters (p1, p4, p14) and mutations (p15, which had a unique *TP53* mutation) relapsed, as compared to 3/8 (37.5%) patients presenting different clusters in pre- and post-NAC samples (Appendix A).

Globally, genes in post-NAC samples were involved in pathways associated with actionable targets in tumor treatment where, beside the canonical PI3K/Akt/mTOR, EGFR, and Ras signaling pathways the most represented terms were related to regulation of cell cycle processes (Table 2).

We finally asked whether in patients not attaining pCR, mutated genes were related to each other in pre-NAC samples. *KMT2A*, *MYH11*, *NOTCH1*, *SYNE1*, *LRP1B*, *FN1*, *BCL9* were identified as significantly co-occurring mutated genes (FDR < 0.05). Oppositely, post-NAC samples showed no co-occurrence. A network of physical interaction between co-occurrent genes identified a module of 246 nodes (Figure 4A) involved in mismatch repair, PI3K/Akt, ERBBB, Ras, Notch, androgen receptor signaling, and regulation of epithelial to mesenchymal transition (Figure 4B). Notably, *NOTCH1* was involved beside Notch signaling also in regulation of epithelial to mesenchymal transition, suggesting a possible relation between the two pathways (Figure 4B).

### 2.5. Tumor-Infiltrating Lymphocytes

In consideration of the finding of a trend of association between unresponsive cases and alterations in the pathways related to immune response, we integrated the genomic analyses with the evaluation of stromal Tumor-Infiltrating Lymphocytes (sTILs). Overall, sTILs were evaluable at baseline, after treatment, and in matched pre- and post-NAC samples in 16, 15, and 11 cases, respectively. sTILs score was 12.6% (range 3–30%) at baseline (Appendix A). There was no significant association between the number of mutations and sTILs (median sTILs in pre- and post-NAC samples 10% vs. 5%, *p* = 0.57; median sTILs in patients with different clusters as compared with those with stable clusters in pre- and post-NAC samples 5% vs. 7.5%, *p* = 0.97). Among 11 evaluable matched pre- and post -NAC samples, increased, decreased and unchanged sTILs were observed in five, four, and two cases, respectively. At baseline primary tumors with stable and mutated mutational profile shared the same mean sTILs values, i.e., 13% (range 5–25%, and range 1–35%), respectively. After treatment with NAC, sTILs remained the same, i.e., 13% (range 3–30%) in stable cases whereas they reached an average value of 19% (range 5–50%) in cases with a modified mutational profile.

## 3. Discussion

The mutational profile of breast cancer has been widely studied in multiple contexts [15,16]. This study conducted a focused assessment of mutational changes in primary TNBC exposed to standard NAC, matched pairs in 63% of cases, over a limited time period, providing an opportunity to isolate anthracycline and taxane associated genomic changes in this clinical context. Our primary results include the finding that TNBC is a very heterogeneous disease with few recurrent mutations. The unique exception was represented by *TP53* which was found altered in 87% of pre-NAC samples, a finding in line with prior studies, including the TCGA dataset [15,16]. In fact, we were not able to find specific tumor somatic mutations associated with response/resistance to NAC. Nevertheless, albeit the sample size, by using external knowledge, i.e., canonical pathways from MSigDB, none of the patients attaining pCR presented alterations in the immune related processes. As known, increasing evidence supports the role of environmental factors in the escape of the tumor cells from the effect of cytotoxic agents. In particular, the influence of innate and adaptive immune responses on the efficacy of chemotherapy has been intensely examined and recently reviewed [17]. Tumor associated macrophages have been documented to contribute to doxorubicin and paclitaxel resistance by direct cathepsin protease B- and S-mediated effects; in addition, their increased influx into tumors leads to a surge of regulatory T and B cells that is accompanied by impaired recruitment of cytotoxic T cells [18]. On the other hand, several chemotherapeutic agents, including doxorubicin, are able to induce immunogenic cell death, antigen-presenting ability of dendritic cells, and a subsequent T cell response [19]. Our analysis revealed that TNBC tumors bearing proficient genes involved in the immune pathways are much more sensitive to chemotherapy. Although our findings require confirmation in larger datasets, evaluation of additional variables including disease, diet, and lifestyle, preferably originating from prospective clinical studies, they provide the impetus for using a target gene profile to aid an effective strategy with near-term clinical impact on TNBC management, by identifying those patients who might benefit from NAC and directing the remaining patients to novel therapeutic strategies including immunotherapy, based on the profile of residual disease.

As known, the FDA has recently approved atezolizumab as an immunotherapy for TNBC treatment (https://www.cancer.gov/news-events/cancer-currents-blog/2019/atezolizumab-triple-negative-breast-cancer-fda-approval), here we provide evidence of other pathways of the immune system that could also serve as targets for novel TNBC therapies. Notably, based on our analysis of deregulated immune response in unresponsive patients, we point out that deregulated NF-kappa-B signaling is known to be implicated in the pathogenesis of numerous human inflammatory disorders and malignancies. Consequently, the NF-kappa-B pathway has attracted attention as a promising therapeutic target for drug discovery, and different compounds have been characterized, with many demonstrating promising efficacy in pre-clinical models of cancer and inflammatory disease (reviewed in [20]). Finally, treatments that increase RNase L activity directly or indirectly are predicted to increase the antitumor activities of well-known anticancer compounds, including 5-azacytidine [21].

We further reported that the overall number of tumor somatic mutations was not increased after NAC in matched post-treatment samples despite DNA-damaging therapy decreased by NAC in matched pre- and post-treatment samples, in most cases. When considering previous studies [15,16], our findings reinforce the concept that mutation changes may be related to the different chemosensitivity levels of mutant and wild-type cancer cells. For patients who had loss of mutations after NAC, it is likely that their cancer cells with certain mutations might be sensitive to chemotherapeutic agents. Therefore, cytotoxic therapy would primarily remove the mutant cells and consequently shift the evolutionary landscape in favor of the non-mutant subclones. Conversely, for patients without loss of mutations after NAC, cancer cells might be resistant to chemotherapeutic agents and in the residual tumors, the mutant alleles remain or become enriched. We further observed tumor somatic mutations private to pre- and post-NAC samples and found that they are primarily sub-clonal. Hence the overall effect of NAC is the reduction of the total number of mutations and the appearance of new mutations not detected in pre-treatment samples. While this finding may be consistent with NAC-induced subclone reduction and selection of pre-existing clones with low rate mutations, the confounding effects of tumor spatial heterogeneity, as observed in multi-regional biopsies, cannot be completely ruled out with this analysis alone [22]. By hierarchical clustering, we found that all patients with concordant pre-and post-NAC clusters or mutations (such as in the case of p15 which presented a unique *TP53* mutation before and after therapy) relapsed as compared to 37.5% of those with NAC-induced changes. Prior studies, reviewed in [23], found that intratumoral heterogeneity (variably defined) was associated with survival and response to therapy. While this may reflect intrinsic tumor biology, our results suggest that stable post-treatment mutational profile (perhaps reflecting broad tolerance to treatment) may provide additional prognostic information; thereof, both pre- and post-treatment assessments of intratumoral heterogeneity may aid in risk stratification and should be explicitly assessed in future clinical trials. In addition, albeit the small sample size, basal and post treatment sTILs appear to be higher in tumors with changed mutational profile upon NAC, a result which is consistent with a possible anti-tumor effect of chemotherapy through modulation of the immune system [24,25]. Additional studies examining the tumor immunological landscape and response to chemotherapy through different “omics” in pre-and post-treatment settings may clarify this relation further. Recently, TNBC has been classified by gene expression into four different subtypes by taking into consideration the contribution of transcripts from normal stromal and immune cells in the tumor environment [26]. These four TNBC subtypes differed in their response to standard neoadjuvant chemotherapy, with basal tumors having a better response than non-basal. Noteworthy, the luminal androgen receptor (LAR) subtype, which indeed presents increased AR signaling, had better survival despite a decreased response to neoadjuvant chemotherapy. Although few patients shared similar tumor mutational profiles, common pathways in unresponsive cases included those of regulation of cell cycle progress, PI3K/Akt and mTOR signaling, EGFR tyrosine kinase inhibitor resistance. Network analysis revealed connectivity among somatic variants in unresponsive cases that defined highly connected modules including PI3K/Akt (KEGG-hsa04151), ERBB (KEGG-hsa04012), Ras (KEGG-hsa04014), Notch (KEGG-hsa04330), androgen receptor signaling pathways (GO:0030521), and mismatch repair (KEGG-hsa03430). Altogether, these results provide support to the concept that despite the lack of common mutations in patients not attaining pCR, the mutations fell into several and druggable shared functional categories.

Our study has also several limitations. The power of analysis was compromised because of small capacity and number of events. Moreover, it suffered the lack of sufficient pre-NAC tumor tissue for microdissection and mutation analysis to address the issue of intratumor spatial heterogeneity. Furthermore, additional studies are warranted to explore the mechanisms resulting in NAC-induced mutational changes, the functional evaluation of emerging mutations involved in potential resistance, as well as proliferation advantage, and the value of immune response to conventional chemotherapy. Importantly, the costs of serial sequencing assays preclude its clinical implication, which might be overcome with technology improvement. Given that targeted gene sequencing dynamics might serve as a potential marker of chemo-sensitivity, its clinical value should be further validated in larger series of patients with TNBC.

## 4. Materials and Methods

### 4.1. Patient Cohort

Pre-treatment and surgically resected tumor samples were from TNBC patients treated with anthracycline/taxane-based NAC and breast and axillary surgery at Fondazione IRCCS Istituto Nazionale dei Tumori di Milano, Italy. Clinical and pathological data were retrieved from the prospectively maintained pathology-based institutional registry [27]. Tumor stage was assigned according to the 7th edition of the TNM manual [28]. Hormone receptor status was evaluated according to the American Society of Clinical Oncology guidelines [29], and ER and PgR negativity was defined as immunohistochemical staining of fewer than 1% of tumor cell nuclei. HER2 status was considered negative when the IHC score was 0–1, or 2+ with a negative chromogenic in situ hybridization result [30]. Stromal Tumor-Infiltrating Lymphocytes (sTILs) were assessed on a single hematoxylin-eosin stained slide, and scored according to pre-defined criteria [31]. All cases were independently evaluated by two experienced BC pathologists (GP and AV) blinded to treatment data and follow up. Breast surgery was performed after a maximum of 6 weeks from the conclusion of NAC. pCR was defined as the absence of invasive BC in the breast and axillary nodes [32]. Collection, use, and analysis of all specimens in this study were approved by the institutional review board and ethics committee (Code D295938), and all patients signed an informed consent form to allow the institutional research on tumor leftover material after the completion of diagnostic procedures.

### 4.2. Targeted Next Generation Sequencing

Archival tissue in formalin-fixed paraffin-embedded (FFPE) macro-dissected tumor obtained during routine diagnostic biopsy and/or resection of primary tumor was used. The tumor content on hematoxylin and eosin-stained slides for each study sample was evaluated by two pathologists (GP and AV) to assess tumor cellularity. In patients attaining pCR only pre-NAC primary tumor samples were analyzed by targeted next generation sequencing (NGS); in those with residual disease at surgery (RD) available pre- and post-NAC tumor samples were both sequenced. For all patients matched germline DNA was obtained from negative lymph nodes or breast normal tissue. DNA was extracted using GeneRead DNA FFPE kit (Quiagen, Hilden, Germany. Tumor and normal samples were analyzed by NGS using the Ion AmpliSeq Comprehensive Cancer Panel (ThermoFisher, Waltham, MA, USA), which cover all exons of 409 cancer-related genes. A total of 40 ng of genomic DNA were used to amplify the 409 all-exon genes using the Ion AmpliSeq Library Kit2.0 (Life Technologies, Carlsbad, CA, USA) according to the manufacturer’s manual (MAN0006735 rev 5.0). Amplicons were ligated to P1 and Barcode adapters using DNA Ligase. Barcoded libraries were purified using AMPure Beads XP (Beckman Coulter, Pasadena, CA, USA) and PCR-amplified for a total of five cycles. After a second round of purification with AMPure Beads, the amplified libraries were sized, their quality was assessed using the Agilent BioAnalyzer DNA High Sensitivity kit (Agilent Technologies, Santa Clara, CA, USA), and DNA was quantified using the Qubit dsDNA HS kit (Life Technologies). In particular, due to the FFPE origin of the DNA, dsDNA fragments with a length range of 50–150 bp were selected. Emulsion PCR and sample enrichment were performed using the Ion One touch 2 instrument according to the manufacturer’s instruction. An input concentration of DNA library obtained with the first amplification step was added to the emulsion PCR master mix and the ion sphere particles (ISPs) and a double phase (oil/water) PCR was performed. Then ISPs were recovered, and template positive ISPs were enriched using Dynabeads MyOne Streptavidin C1 beads (Life Technologies). Libraries were finally sequenced using the Ion Personal Genome Machine (PGM) platform using ION PGM 200 sequencing kit V2 according to the manufacturer’s instructions.

### 4.3. Sequencing and Bio-Informatic Data Analysis

Raw sequencing data were mapped to the human reference genome (hg19) using the TMAP algorithm implemented in the Torrent Suite software version 4.4.3 with default parameters. Aligned bam files for matched tumor and normal samples were analyzed for somatic variant calling using the AmpliSeq CCP w1.1. Tumor-Normal pair workflow of Ion Reporter version 5.10. Mutations were annotated using Oncotator v1.9.9.0 [33]. Finally, to reduce the number of false positive calls, and obtain a list of confident somatic mutations we discarded variants passing at least one of the following filters: (i) phred quality score < 30; (ii) strand bias *p*-value < 0.01; (iii) number of variant-supporting reads < 7; (iv) variant allelic frequency (VAF) < 5% and depth < 500×; (v) VAF < 10% and C > T/G > A substitution (possible FFPE artifacts); (vi) variant presence in dbSNP or depth in normal sample < 10× and allele frequency ≥ 1% in the 1000 Genomes Project European population or ExAC non-Finnish European population; (vii) depth in normal sample < 10×; (viii) variant allelic frequency (VAF) in normal sample ≥ 10%. Somatic mutations, identified with Mutect2 algorithm, and clinical data for breast cancer patients of TCGA database were downloaded from the Genomic Data Commons data portal (https://portal.gdc.cancer.gov/) with accession date 15 May 2019. Matched mutational and clinical data were available for 986 patients and among them 100 TNBC patients (10.1%) were selected according to the negative ER, PgR, and HER2 status, as defined by immunohistochemistry. None of these TNBC cases received neo-adjuvant therapy while information about adjuvant therapy was not available for the majority of patients (*n* = 83). Only four patients received adjuvant radiotherapy and 12 received adjuvant pharmacological treatment. Mutational analysis was performed using the maftools package and was restricted to the 409 genes included in the Ion AmpliSeq Comprehensive Cancer Panel.

### 4.4. Statistical Analysis for Association of Genomic and Clinical Data

The maftools Bioconductor package was used to summarize, analyze, and visualize somatic variants [34]. Only variants assumed to have high or moderate impact in the protein, probably causing protein truncation, loss of function, or triggering nonsense mediated decay were included in downstream analyses, except tumor mutational burden calculation, where we considered both high/moderate impact mutation or all somatic mutations. To identify mutations associated to NAC response we encoded the mutational status as a binary variable, corresponding to 0 if the gene had no mutations in a given sample and 1 if the gene had at least one mutation, and we applied Fisher’s exact tests. Analysis at the pathway level was performed using the C2 canonical pathways gene sets retrieved from the Molecular Signatures database (http://software.broadinstitute.org/gsea/msigdb). For each sample we classified each pathway as mutated if at least one of its genes carried a mutation. We then associated mutated pathways to NAC response using Fisher’s exact test. For paired pre- and post-NAC analysis, clustering of genes according to VAF was performed using unsupervised hierarchical clustering with Ward linkage and Euclidean distance. Co-occurring mutated genes were identified using the *somaticInteractions* function of maftools package (*p*-value < 0.05). Biogrid Human protein-protein interaction (PPI) network, Igraph [35], and rTRM [36] R packages were used to generate connected graphs between co-occurring mutated genes. Functional enrichment of gene lists for using Gene Ontology (GO) biological process terms and KEGG pathways was performed using the ClusterProfiler Bioconductor package. Enrichments with a false discovery rate (FDR) < 0.05 or < 0.01 were considered for KEGG pathways or GO terms, respectively.

Association between continuous and categorical variables was assessed using Wilcoxon rank-sum test for unpaired comparisons and Wilcoxon signed-rank test for paired analyses. All statistical tests were two-sided and associations were considered significant when the *p*-value was < 0.05.

## 5. Conclusions

Targeted gene sequencing provides insights into the clinical and genomic dynamics of tumor evolution in TNBC patients treated with NAC, suggesting mechanisms of clinical resistance, and informing the development of clinically relevant biomarkers and trials of combination therapy. Specifically, NAC may affect somatic mutation status in patients with TNBC, lowering the number of mutations. This may be attributable to the heterogeneity of intratumoral mutations and the different sensitivities of mutant and wild-type tumor cells to chemotherapy. The immune system is involved in response to NAC as unresponsive cases are enriched of mutations impinging on immune processes and present less TILs. Finally, post-NAC samples are enriched of mutations impinging on cell cycle and growth factor signaling which are exploitable with specific targeted agents. These findings might help to optimize the choice of sequential therapy and improve patient survival. To enable the selection of the most effective therapeutic option, additional clinical studies are needed to further examine drug resistance mechanisms and to validate our findings in other independent cohorts. We anticipate that we will undertake a prospective neoadjuvant trial designed to analyze primary tumor mutational status and treatment activity. In this trial, early TNBC patients will be treated with anthracycle and taxane-based NAC. This trial will recruit patients from multiple centers in Italy, under the umbrella of the Agency of the Italian Ministry of Health Alleanza Contro il Cancro.

## Figures and Tables

**Figure 1 cancers-11-01753-f001:**
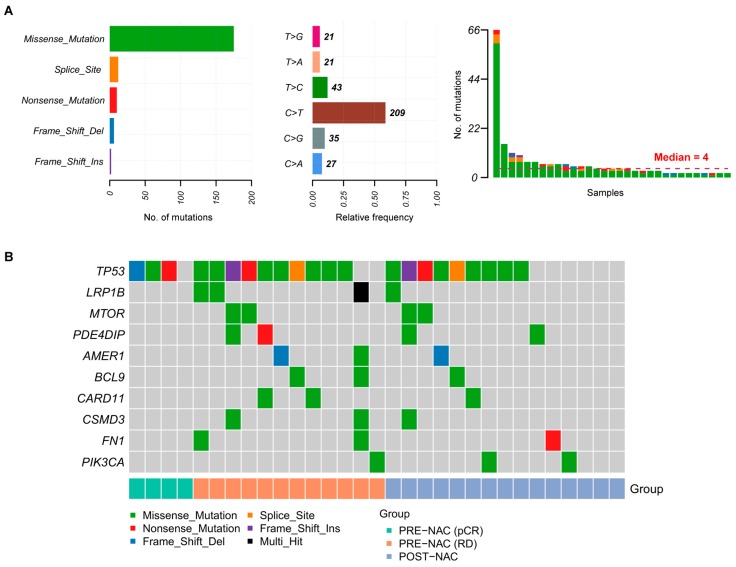
Mutational spectrum of triple negative breast cancer (TNBC). (**A**) Bar plots showing the number of somatic mutations by type, the relative frequency of nucleotide substitutions, and the number of mutations per sample colored by mutation type. (**B**) Oncoplot reporting the top-10 most recurrently mutated genes across the TNBC samples analyzed in this study. Samples are ordered according to response group. Genes are listed from top to bottom by decreasing frequency.

**Figure 2 cancers-11-01753-f002:**
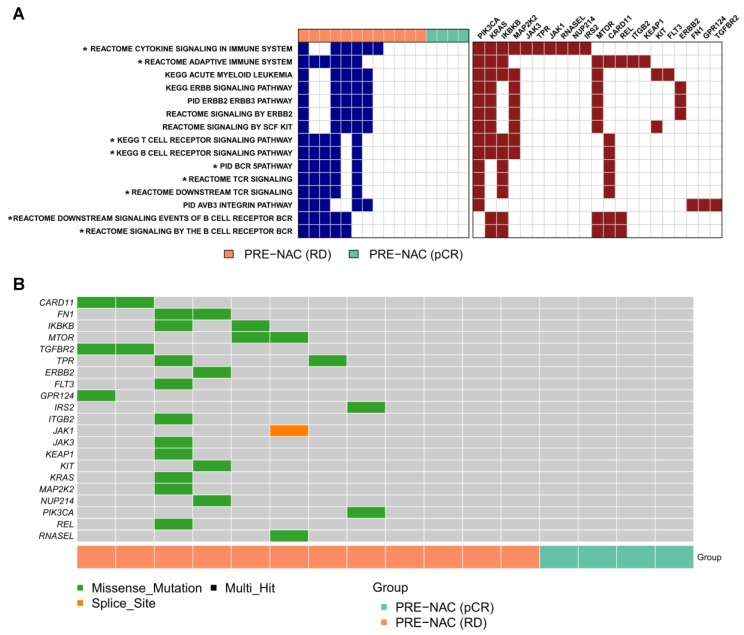
Mutated pathways associated to pathological Complete Response (pCR) in pre-neo-adjuvant chemotherapy (NAC) tumors. (**A**) Heatmap showing the 15 pathways (rows) that are preferentially mutated in residual disease (RD) patients compared to pCR ones (columns). Asterisks indicate immune-related pathways. Blue squares indicate that the pathway is mutated in the sample. Mutated genes (red squares) included in each pathway are reported. (**B**) Oncoplot showing the mutational status of the genes belonging to the 15 pathways associated to NAC response across pre-NAC samples.

**Figure 3 cancers-11-01753-f003:**
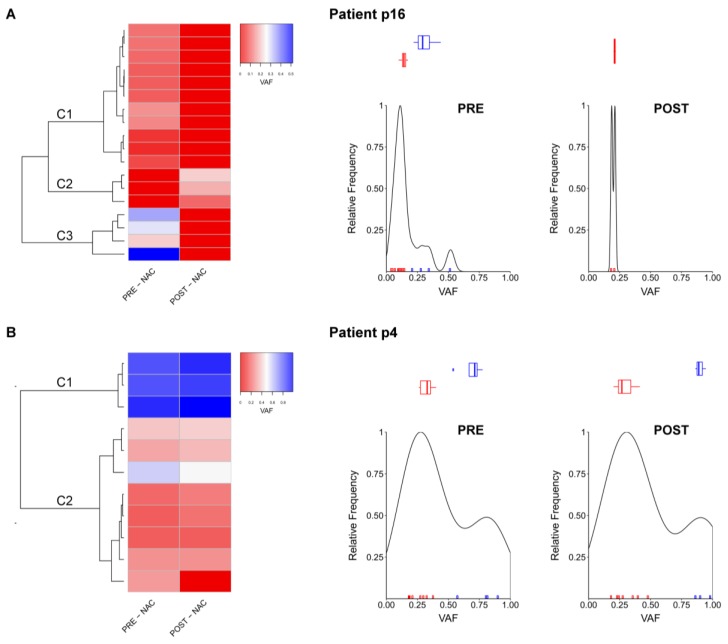
Evaluation of mutant allele distribution between matched pre- and post-NAC samples. Non-synonymous mutations of representative patients with residual mutations in post-NAC samples (**A**) and without evidence of change from post-NAC samples (**B**) are grouped and represented in red-blue scale according to their frequency.

**Figure 4 cancers-11-01753-f004:**
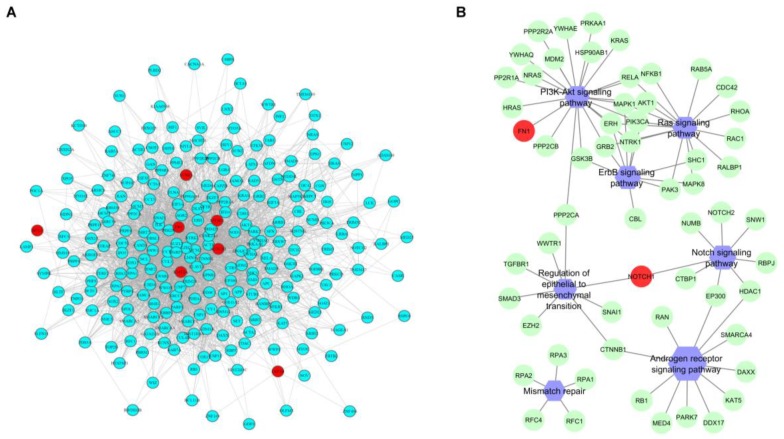
Functional interactions network analysis of co-occurring mutations in pre-NAC samples. (**A**) A network of 246 nodes were obtained starting from co-occurred mutated genes (highlighted in red) using Biogrid protein-protein interaction data. (**B**) Most representative targeted signaling pathway (represented in violet) were obtained from Gene Ontology (GO) and Kyoto Encyclopedia of Genes and Genomes (KEGG) databases.

**Table 1 cancers-11-01753-t001:** Patient, primary tumor, and treatment characteristics.

Characteristics	N (%)
**Patient age**	
<50 years	12 (63)
≥50 years	7 (37)
**Clinical T size**	
2–5 cm	13 (68)
>5 cm	6 (32)
**Clinical nodal status**	
N0	8 (42)
N1-3	11 (58)
**Clinical stage**	
IIA	5 (26)
IIB	10 (53)
IIIA	2 (11)
IIIB	1 (5)
IIIC *	1 (5)
**Tumor grade**	
G2	1 (5)
G3	18 (95)
**Ki67**	
<30%	3 (16)
≥30% <60%	3 (16)
≥60%	12 (63)
Missing	1 (5)
**Type of NAC**	
– Doxorubicin/Paclitaxel every 3 weeks followed by CMF1–8 every 4 weeks ^	11 (58)
– Doxorubicin/Paclitaxel every 3 weeks followed by Eribulin 1–8 every 3 weeks ^^	5 (26)
– Other	3 (16)
**Path Findings**	
**pCR**	
ypT0N0	4 (21)
**Residual Disease**	
ypT1N0	6 (32)
ypT2-3N0	7 (37)
ypT2N1	1 (5)
ypT4N3	1 (5)
**Evaluable Cases**	
Pre-NAC	16 (84)
Post-NAC	15 (79)
Paired pre- and post-NAC	12 (63)
**Events**	
Distant metastases	7 (37)

^ Gianni et al. Clin Cancer Res 2005; ^^ Di Cosimo et al. PlosOne 2019; * suspected bone metastases at initial presentation, not histologically proven.

**Table 2 cancers-11-01753-t002:** Pathway enrichment analysis of mutations in patients with changed mutational profile between pre- and post-NAC samples.

Patient	Cluster	No. of Genes in Cluster	No. of Genes in Cluster Mutated in POST-NAC Tumor	Enriched Pathways Including Genes Mutated in POST-NAC Tumor
p5	C1	42	1	No pathways found
	C2	10	1	No pathways found
	C3	15	0	NA
p10	C1	3	3	EGFR tyrosine kinase inhibitor resistance (KEGG - hsa01521); Ras signaling pathway (KEGG - hsa04014)
	C2	3	0	NA
p11	C1	3	2	Negative regulation of cell cycle process (GO:0010948); Androgen receptor signaling pathway (GO:0030521)
	C2	5	2	Ras signaling pathway (hsa04014); TOR signaling (GO:0031929); EGFR tyrosine kinase inhibitor resistance (KEGG - hsa01521)
p12	C1	3	3	mTOR signaling pathway (KEGG - hsa04150); PI3K-Akt signaling pathway (KEGG - hsa04151);
	C2	2	0	NA
p13	C1	1	1	PI3K-Akt signaling pathway (KEGG - hsa04151); Negative regulation of cell cycle process (GO:0010948)
	C2	4	2	TORC1 signaling (GO:0038202)
p16	C1	11	0	NA
	C2	3	3	Hippo signaling pathway (KEGG - hsa04390)
	C3	4	0	NA
p18	C1	2	2	Regulation of cell cycle arrest (GO:0071156); PI3K-Akt signaling pathway (KEGG - hsa04151)
	C2	2	2	Mismatch repair (KEGG - hsa03430); Platinum drug resistance (KEGG - hsa01524)
	C3	2	0	NA

NA: not applicable.

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
