# Peer review of "Targeted-Gene Sequencing to Catch Triple Negative Breast Cancer Heterogeneity before and after Neoadjuvant Chemotherapy"

_cancers, 2019, doi:10.3390/cancers11111753_

Round 1
Reviewer 1 Report
All suggestions have been addressed by the authors now.
Congratulations for this piece of work to be published by CANCERS.
With kind regards
Author Response
We thank Reviewer#1 for congratulating us.
Reviewer 2 Report
This reviewer really appreciates the authors’ efforts to answer this reviewer’s comments. The revised version of the manuscript improves the clarity and quality of the article.
However, this reviewer still has some concerns.
In the response to this reviewer comment #1, the authors included a most accurate definition of triple negative breast cancer (TNBC). In the response to this reviewer the authors indicated: “at page 1, line 37, ie “Triple-negative breast cancer (TNBC), defined by the absence of estrogen (ER), progesterone (PR), and overexpression and/or gene amplification of HER2 receptor”. However, in the manuscript the authors stated: “Triple-negative breast cancer (TNBC), defined by the absence of estrogen (ER), progesterone (PR), and overexpression and/or amplification of HER2 receptors”. Please include in the manuscript that TNBC do not present HER2 overexpression or HER2 GENE In response to Comment #7 from this reviewer, the authors included data in Figure S2. This reviewer also suggests to include the reference to Figure S2 in the manuscript page 4, line 101 after the sentence: “A trend toward decreased number of mutations was observed in patients = 50 compared to those >50 years old, 4 (2-6) versus 7 (5-66), p=0.002”. This reviewer would like to congratulate the authors for the changes provided in the results and discussion sections related to comment #9. They provided a great description of the results related with genes of the immune system. They also discussed the therapeutic utility on their results based on the literature. However, this reviewer still believes that the authors should mention that the FDA has approved an immunotherapy for TNBC treatment (atezolizumab) https://www.cancer.gov/news-events/cancer-currents-blog/2019/atezolizumab-triple-negative-breast-cancer-fda-approval; https://www.nejm.org/doi/full/10.1056/NEJMoa1809615. Interestingly, this study provides evidences of other pathways of the immune system that could also serve as targets for novel therapies.
Minor concerns:
1- In response to comment #10 (page 1, line 26), this reviewer still noted that there is a space missing after pre- and before the word “and”, this has not been fixed in the manuscript provided to this reviewer.
Author Response
In the response to this reviewer comment #1, the authors included a most accurate definition of triple negative breast cancer (TNBC). In the response to this reviewer the authors indicated: “at page 1, line 37, ie “Triple-negative breast cancer (TNBC), defined by the absence of estrogen (ER), progesterone (PR), and overexpression and/or gene amplification of HER2 receptor”. However, in the manuscript the authors stated: “Triple-negative breast cancer (TNBC), defined by the absence of estrogen (ER), progesterone (PR), and overexpression and/or amplification of HER2 receptors”. Please include in the manuscript that TNBC do not present HER2 overexpression or HER2 GENE
In reply
We thank Reviewer #2, and we modified the definition of triple negative breast cancer according to his/her suggestions at pag , line 37.
In response to Comment #7 from this reviewer, the authors included data in Figure S2. This reviewer also suggests to include the reference to Figure S2 in the manuscript page 4, line 101 after the sentence: “A trend toward decreased number of mutations was observed in patients = 50 compared to those >50 years old, 4 (2-6) versus 7 (5-66), p=0.002”.
In reply
We thank Reviewer #2, and we included the reference to Figure S2, as indicated.
This reviewer would like to congratulate the authors for the changes provided in the results and discussion sections related to comment #9. They provided a great description of the results related with genes of the immune system. They also discussed the therapeutic utility on their results based on the literature. However, this reviewer still believes that the authors should mention that the FDA has approved an immunotherapy for TNBC treatment (atezolizumab) https://www.cancer.gov/news-events/cancer-currents-blog/2019/atezolizumab-triple-negative-breast-cancer-fda-approval; https://www.nejm.org/doi/full/10.1056/NEJMoa1809615. Interestingly, this study provides evidences of other pathways of the immune system that could also serve as targets for novel therapies.
In reply
We thank Reviewer #2 for this comment, and accordingly we introduced the suggestion at page 10, line 223.
Reviewer 3 Report
A major concern with the present study is the small number of patient, especially since these are divided into different sub groups that also show heterogeneity. Consequently, it is difficult to evaluate statements about there being no significant differences when the sample size is so small. All statements like “a trend towards decreased number” (line 99) also lacks impact since the n value is small for human subjects with variable disease, diets and life styles. The over use of acronyms, particulary in the abstract, makes reading this paper unduly difficult for someone not used to the non-standard abbreviations.
Author Response
A major concern with the present study is the small number of patient, especially since these are divided into different sub groups that also show heterogeneity. Consequently, it is difficult to evaluate statements about there being no significant differences when the sample size is so small. All statements like “a trend towards decreased number” (line 99) also lacks impact since the n value is small for human subjects with variable disease, diets and life styles.
In reply
We acknowledge the Reviewer #3 comment, and accordingly:
We just reported the observation on mutations according to age, eliminating the wording “a trend toward…” In the main text the statement “no significant differences” reported just in one occasion (at page 6, line 137 when tumor mutational load between pre- (n=16) and post-NAC (n=15) samples was compared, has been replaced by “no differences” . Finally, we added the Reviewer #3 suggestion in the paragraph at page 10, line 217 as following: “Although our findings require confirmation in larger datasets, evaluation of additional variables including disease, diet and life style, preferably originating from prospective clinical studies, they provide the impetus for using target gene profile to aid an effective strategy with near-term clinical impact on TNBC management, by identifying those patients who might benefit from NAC and directing the remaining patients to novel therapeutic strategies including immunotherapy, based on the profile of residual disease.”
The over use of acronyms, particulary in the abstract, makes reading this paper unduly difficult for someone not used to the non-standard abbreviations.
In reply
We reduced the use of acronyms in the abstract, accordingly.
Round 2
Reviewer 2 Report
This reviewer really appreciates the authors’ efforts to answer this reviewer’s comments. The revised version of the manuscript improves the clarity and quality of the article. The authors have adequately responded to all the concerns.
Reviewer 3 Report
The authors have addressed my comments apart from the problems of a small sample size.
This manuscript is a resubmission of an earlier submission. The following is a list of the peer review reports and author responses from that submission.
Round 1
Reviewer 1 Report
Congratulations to this groundbreaking work on targeted gene sequencing in triple negative breast cancer. A heterogenous disease is explored with high effort in neoadjuvant and post-neoadjuvant breast cancer samples and the genomic changes are monitored specifically to identify targets for further treatment in residual disease which indicates chemoresistance and is always associated with a poor prognosis.
Some questions occur to this manuscript however:
The sample size is low and includes also a case with suspected bone metastases. In metastatic disease, other drivers might be activated different from the neoadjuvant and adjuvant setting. The pCR-rate is comparably low at 21 %. This is significantly lower than the pCR-rate of the MD Anderson cohort treated with anthracycline and taxane and even much lower than in the patients treated with anthracycline, taxane and carboplatin in the GeparSixto study or our own studies. The dose and schedule (AC/EC: 3-weekly? 2-weekly?, Taxane: weekly?) should be specified in the manuscript. The percentage of availability of matched pairs of pre- and post-chemo specimen, which was 63 %, should be mentioned in the discussion. "p" should be clarified as an abbreviation of "patient", otherwise this could be misunderstood by the reader. "mensenchimal" in the manuscript should be changed to "mesenchymal" (line 156 & 157) the "comma" before "in stable cases" should be removed (line 174)
We recommend that you include Carboplatin in future studies as it significantly demonstrated to improve the pCR-rate and subsequently disease free survival by almost 10 % in GeparSixto trial (Loibl S, Ann Oncol 2018 Dec 1; 29 (12):2341-2347) and in our own studies compared with non-platin-containing regimen. As published in by Hahnen E (JAMA Oncol. 2017 Oct 1;3(10):1378-1385) this benefit is most pronounced in the non-BRCA mutated patients.
With best regards!
Reviewer 2 Report
The authors used targeted-gene sequencing to catch triple negative breast cancer heterogeneity before and after neo-adjuvant chemotherapy. Since TNBC patients not attaining pathological Complete Response (pCR) after neo-adjuvant chemotherapy (NAC) have poor prognosis, the authors investigated somatic mutations in 409 cancer-related genes in primary tumor biopsies and residual disease from 19 TNBC patients treated with NAC. This review is of high interest for clinicians specialized in treatment of breast cancer worldwide. This article's findings may support the use of targeted-gene sequencing to predict the response to NAC as well as to reveal novel actionable targets in the residual disease.
This referee has some concerns that could improve the clarity and quality of the manuscript.
Here follows the list of concerns:
In the "Introduction" section (page 1, lane 36), triple negative breast cancer is defined as: “Triple-negative breast cancer (TNBC), defined by the absence of estrogen (ER),progesterone (PR), and HER2 receptors, accounts for approximately 15% of all BC cases”. Indeed, TNBC may express HER2 receptor but in low levels, which are not considered as HER2 overexpression. Therefore, a most accurate definition of TNBC indicates that these tumors lack ER and PR expression and do not present HER2 overexpression or HER2 gene amplification.
It is currently well-acknowledged that TNBC is a heterogeneous disease, which results from a diagnosis of exclusion. Indeed, TNBC are currently classified into 4 different molecular subtypes (TNBC-type 4 classification, Pietenpol and coworkers, https://doi.org/10.1371/journal.pone.0157368). This reviewer believes that this classification should be explained along the manuscript and also encourage the authors to discuss their results in the light of the current molecular TNBC classification and known heterogeneity. Interestingly, post-NAC samples showed mutations in androgen receptor signaling pathways among different features. One of the molecular subtypes of TNBC is the luminal androgen receptor (LAR) subtype, which indeed presents increased AR signaling.
This reviewer suggests to include more references to the figures and tables within results section of the manuscript. Particularly in results section 2.1, 2.3 and 2.5, this reviewer noted that many of the results described were not referred to any figure or table, thus making it a bit difficult to the reader to follow the article. For example:
Page 2, line 68: The authors describe the percentage of patients with different clinical stages. This reviewer could not find any reference to the figure or table where this information is depicted. This information is missing in Table 1 and Sup. Table 1.
Page 8, line 165: Description of the results regarding sTILs does not present a reference to any figure or table within the article, so the reader doesn't know where to find the results described. This reviewer noted that Supplementary Table 1 contains the % of sTILs corresponding to each sample. The authors also mentioned: “There was no significant association between the number of mutations and sTILs”. Please indicate the table where this information is shown.
This reviewer suggests to include some modifications in Table 1, in order to provide extra information to the reader. Please include the percentage along with the number of patients for each characteristic. Please include the clinical stage for each patient. Please include the number of patients with pCR and RD, as well as the number of events registered for this cohort. Although this information is in Sup. Table 1, this reviewer believes it should be provided in one of the main tables of the article.
In order to validate the findings observed in their cohort, the authors performed bioinformatics analysis of mutational data of TNBC patients from TCGA cohort. Please indicate in “Material and methods” section how this analysis was performed, including: how the data was obtained and processed, how patients from TCGA were classified as TNBC patients, how many patients were classified as TNBC patients and included in the analysis shown, did these patients receive neo-adjuvant or adjuvant chemotherapy. Besides, the authors mentioned (page 4, line 95): “A trend toward decreased number of mutations was observed in patients ≤ 50 compared to those >50 years old, 4 (2-6) versus 7 (5-66), p=0.002”. Please indicate the figure or table where this result is shown.
The authors found that TP53 gene (which encodes for p53 tumor suppressor) was among the most mutated genes among both pre- and post-NAC. Mutations in TP53 are the most common mutations in breast cancer. The mutation frequency varies depending on subtype, and TP53 mutations are present in about 80% of the basal-like subtype (doi:10.1038/nature11412). This reviewer believes this information should be provided to the reader and discussed in the manuscript.
Interestingly, 50% of patients with RD were characterized by alterations in pathways related to adaptive immunity. This reviewer suggests a detailed description of the genes of the immune system that are mutated in these patients. I would also be interesting that the authors discuss which are the genes from the immune system that are mutated and if these mutations have previously been reported in TNBC. Recently, the FDA has approved an immunotherapy for TNBC treatment (atezolizumab). It would be interesting to include this information within the discussion and to comment how the results from this study would help to select patients for treatment with atezolizumab. Were PD-L1 or PD-1 mutated among these patients?. This information could also be mentioned.
Minor concerns:
Page 1, line 25: Please note that there is a space missing after pre- and before the word “and”.
Page 6, line 141: Please note that “3 over 6 patients” should be replaced by “3 over 5 patients”
Supplementary Table 4: When you open the file you will find that the table is named as Supplementary Table 1.
Reviewer 3 Report
Targeted-gene sequencing to catch triple negative breast cancer heterogeneity before and after neoadjuvant chemotherapy.
Cosimo et al. report the variant findings in matched samples pairs (biopsies from primary and residual tumor tissue) from 12 patients with TNBC enrolled in a neoadjuvant chemotherapy (NAC) treatment scheme. The overall aim of characterizing the mutational landscape of TNBC before and after NAC and correlate to prognosis is sound and overall the study is well-written. The authors conclude that the results support the use of targeted-gene-sequencing for future therapeutic development and suggests that patients without a complete pathological response have impaired immune-response. However, there are some major concerns which could weaken the scientific merit.
Points of concern:
The number of patients is very small and makes the results preliminary It does not appear that data are manually curated (e.g. 66 mutations could include a lot of false-positive/technical variants) It does not appear that variants are classified according to standard somatic variant classification, e.g. in PIK3CA, normally, activation mutations would weigh different from an insignificant missense variant Figure 2A diagram does not seem to be curated (many of the listed pathways are redundant, which makes the signature stand out visually) Around 10-20% of all TNBC – patients are BRCA1/2 positive carriers. This could affect the mutational signature and the prognosis but appears not to have not been considered. Figure 4 is not very informative